# Antimicrobial Activity of Silver-Treated Bacteria against Other Multi-Drug Resistant Pathogens in Their Environment

**DOI:** 10.3390/antibiotics9040181

**Published:** 2020-04-15

**Authors:** Doaa Safwat Mohamed, Rehab Mahmoud Abd El-Baky, Tim Sandle, Sahar A. Mandour, Eman Farouk Ahmed

**Affiliations:** 1Microbiology and Immunology Department, Faculty of Pharmacy, Deraya University, Minia 11566, Egypt; dd_25_sfwt@yahoo.com (D.S.M.); sahar.mandour@deraya.edu.eg (S.A.M.); dremanfarouk7@gmail.com (E.F.A.); 2Microbiology and Immunology Department, Faculty of Pharmacy, Minia University, Minia 61519, Egypt; 3School of Health Sciences, Division of Pharmacy & Optometry, University of Manchester, Manchester M13 9NT, UK; timsandle@btinternet.com

**Keywords:** *E. coli O104:H4*, sustained release, silver nanoparticles, polluted water and burn dressing

## Abstract

Silver is a potent antimicrobial agent against a variety of microorganisms and once the element has entered the bacterial cell, it accumulates as silver nanoparticles with large surface area causing cell death. At the same time, the bacterial cell becomes a reservoir for silver. This study aims to test the microcidal effect of silver-killed *E. coli* O104: H4 and its supernatant against fresh viable cells of the same bacterium and some other species, including *E. coli* O157: H7, Multidrug Resistant (MDR) *Pseudomonas aeruginosa* and Methicillin Resistant *Staphylococcus aureus* (MRSA). Silver-killed bacteria were examined by Transmission Electron Microscopy (TEM). Agar well diffusion assay was used to test the antimicrobial efficacy and durability of both pellet suspension and supernatant of silver-killed *E. coli* O104:H4 against other bacteria. Both silver-killed bacteria and supernatant showed prolonged antimicrobial activity against the tested strains that extended to 40 days. The presence of adsorbed silver nanoparticles on the bacterial cell and inside the cells was verified by TEM. Silver-killed bacteria serve as an efficient sustained release reservoir for exporting the lethal silver cations. This promotes its use as a powerful disinfectant for polluted water and as an effective antibacterial which can be included in wound and burn dressings to overcome the problem of wound contamination.

## 1. Introduction

Increasing bacterial resistance due to inappropriate use of antibiotics is one of the most important problems facing the modern scientific community [1]; moreover, the emergence of new resistant bacterial strains to current antibiotics has become a serious public health issue. This, has raised the need to develop new bactericidal materials [2]. Hence, there is an urgent need for new approaches to overcome antibiotic resistance and to develop alternate antimicrobial agents that can control infectious diseases, the spread of pathogens and which have long-term effectiveness [3]. One area to prolong antimicrobial activity is with the incorporation of antimicrobial agents within sustained-release delivery systems for their continuous use [4,5,6]. The objective of using new sustained-release drug delivery systems is to enhance the therapeutic effectiveness of drugs, by decreasing their side effects and increasing their bioavailability in certain sites. Among the various drug delivery systems, silver nanoparticles have been widely studied, indicating that the slow release of silver cations in trace amounts is toxic to bacteria and achieves prolonged biocidal activity [7,8]. Alternative drug delivery systems include liposomes, alginate based microparticles, and magnetic capsules. Silver nanoparticles have an advantage over other antimicrobial nanoparticles in terms of more effective antimicrobial activity, which is achieved as silver ions are released from the crystalline core which triggers chemical disequilibrium to occur within bacterial cells. In addition, other nanoparticles require the initiation of a photo-thermal effect, which can lead to hypothermia occurring in some patients. Moreover, nanoparticles that are designed to initiate an antibiotic release process can suffer from a variable release process, which affects drug efficacy.

Silver is a safe inorganic antibacterial and it has the ability to kill many types of disease-causing microorganisms [9]. Furthermore, silver is bactericidal at minute concentrations, exhibiting an “oligodynamic” effect through the presence of toxic metal ions [10]. Through this, the activity of silver facilitates its use as antifungal, anti-inflammatory and antibacterial agent [11,12]. The use of silver ions (Ag^+^) and its compounds are used in formulation of dental resin composites, bone cement, ion exchange fibers and coatings for medical devices [13,14]. The use of silver metal as bactericidal agent requires the oxidation to the Ag^+^ ion, which is a slow release process under normal conditions and leads to low effective silver concentrations. Hence silver salts in the form of silver nitrate have been used for medical applications [15]. The ability of silver to ionize in solution is the main property that makes it very effective at combating bacteria as silver will begin to dissolve and ionize when exposed to water, bodily fluids, or organic tissue. An important advantage is that silver ions have a relatively low toxicity to human cells while adversely affecting bacteria and fungi by interacting with bacterial cell membranes inhibiting reproduction of harmful bacteria [16]. Bacterial kill occurs as the ions deposit themselves into the cell walls and vacuoles of bacteria, damaging cell structures including the cell envelope, cytoplasmic membrane, and the membrane’s contents. Once inside the cell, silver ions bind to DNA and RNA molecules, causing them to condense. This makes it more difficult for ribosomes to transcribe or read the DNA and RNA, a process necessary to protein synthesis and cell division [17]. Due to these properties, silver is a practical antimicrobial agent that will be utilized in years to come.

Sustained-release silver products have an on-going bactericidal action, reducing the risk for colonization and preventing infection. Silver nanoparticles (Ag NPs) have also been found to be effective water disinfectants [18]. Conventional water disinfectants, such as free chlorine, may produce harmful disinfection byproducts (DBPs); many of which are considered carcinogens, mutagens, and teratogens [19]. The bactericidal effect of Ag NPs is slow but long-term and persistent, inhibiting microorganisms present in drinking water and avoiding the production of hazardous DBPs [20]. In addition, silver nanoparticles (NPs) carrying and releasing Ag^+^ in a sustained release form may significantly improve the efficacy of wound treatment compared with current therapies. Topical administration of AgNPs allows optimal delivery to the dermis and improves product efficacy [21]. Furthermore, associating NPs with dressings are recent tools for wound healing treatment, especially with regard to their multifunctional properties [22].

What has been less well-researched is the extent that silver-killed bacteria exhibit a prolonged antimicrobial activity against other organisms. In this study, we tested the antimicrobial activity of silver killed *E. coli* O104:H4 against viable population of the same bacterium and some other notable pathogens, such as *E. coli* O157:H7, MDR *P. aeruginosa* and methicillin resistant *S. aureus* (MRSA). *E. coli* O104:H4 is an important water and food-borne pathogen that combines the virulence characteristics of enterohemorrhagic *E. coli* (EHEC) and enteroaggregative *E. coli* (EAEC) and can cause severe diseases and outbreaks. *Ps. aeruginosa* and *S. aureus* were selected because these pathogens represent common causes of wound, burn and nosocomial infectious agents; *E. coli* 0157: H7 was selected as a representative strain of food-borne pathogen (and it is of clinical significance).

## 2. Materials and Methods

### 2.1. Chemicals

AgNO_3_ was purchased from Sigma-Aldrich (St. Louis, MO, USA).

### 2.2. Bacteria

*Escherichia coli* O104:H4, *E. coli* O157:H7, MRSA, and MDR *Ps. aeruginosa* were obtained from the Department of Microbiology, Faculty of Pharmacy, Minia University. All cultures were maintained in their appropriate agar slants at 4 °C and used as stock cultures. This temperature regime ensures that the bacteria used did not grow in number while maintaining cell viability (viability was demonstrated using controls).

### 2.3. Determination of the Minimum Inhibitory Concentration of AgNO_3_

The minimum inhibitory concentration (MIC) of silver nitrate was determined by agar well diffusion assay using AgNO_3_ concentrations 40 ppm, 20 ppm, 10 ppm and 5 ppm with an inoculum of 1 × 10^6^ cfu/mL. MIC values were evaluated after 24 h incubation at 37 °C by measuring zones of inhibition and averaged. The MIC was calculated by plotting the natural logarithm of the concentrations of AgNO_3_ against the square of zones of inhibition. A regression line was drawn through the points. The antilogarithm of the intercept on the logarithm of concentration axis gave the MIC value [23,24].

### 2.4. Evaluating the Antibacterial Effect of Silver-Killed Bacteria

Liquid culture was prepared from *E. coli* O104:H4 isolate by inoculation into nutrient broth and incubation at 37 °C for 24 h. This overnight culture was centrifuged at 4000 rpm for 10 min. The supernatant was discarded, and pellet was washed in pyrogen free water. The washing process by centrifugation was repeated for three times. Pellet was resuspended in 5 mL pyrogen-free water. The suspension was adjusted to 0.5 McFarland. The bacterial suspension (1.0 mL) was added to sterile centrifuge tubes containing 2 mL solution of AgNO_3_ (3.0 mL total volume) at increasing final concentrations of AgNO_3_ (1.5, 3, 6, 12, 18 ppm) and the mixture was incubated overnight at 37 °C under dark conditions for 6 h. Silver-treated bacteria were centrifuged at 4000 rpm for 10 min and the pellet was re-suspended in pyrogen free water. The supernatant was filtered using 0.2 µm syringe filter. Then, 2 mL of the supernatant was added to 1 mL of fresh viable *E. coli* O104:H4 culture and 2 mL of pellet suspension was added to 1 mL of the fresh viable *E. coli* O104:H4 culture. Both supernatant and pellet suspension containing cultures were incubated overnight at 37 °C. Mixtures were serially diluted (10-fold dilution) in saline and pour–plated. The plates were incubated at 37 °C for 24 h, the bacterial colonies were counted and compared to number of colonies of fresh viable *E. coli* O104:H4 culture (not treated) [25].

### 2.5. Heat-Treated Bacteria Control Test

The effect of *E. coli* O104: H4 treated with heat towards viable bacteria was performed by autoclaving the bacterial broth at 121 °C for 15 min, and then 2 mL of autoclaved bacterial broth were added to 1 mL of fresh viable E. coli culture for 24 h [25].

### 2.6. Transmission Electron Micrograph

Silver-killed cells of *E. coli* O104: H4 were centrifuged to separate them from their solution and then re-suspended in 1.0 mL saline. TEM was used to characterize the shape of the silver-killed cells. A drop of bacterial suspension was placed onto carbon-coated copper grid and this was dried in the air to get images for TEM analysis by electron microscope (Hitachi, Japan). All experiments were performed under sterile conditions and in triplicate [26].

### 2.7. Testing Antibacterial Effect of Silver-Killed E.coli O104:H4 Against other Bacterial Species

Agar well diffusion assay was used to test the effect of both silver-killed cells and its supernatant against *E. coli* O157:H7, MRSA, and MDR *Ps. aeruginosa* isolates.

### 2.8. In-Vitro Time Kill Assay

Fresh viable *E. coli O104:H4* was used to inoculate tubes containing 8 mL of MHB with silver nitrate solution of a concentration of 6 ppm, pellet suspension and supernatant. A tube with MHB alone was inoculated as a control. Tubes were incubated at 37 °C with gentle shaking for defined times (0,1, 15, 30, 60, 90 and 120 min). One ml of bacterial suspension was withdrawn and serially diluted in MHB. Twenty-five microliters of each dilution were spotted on Mueller Hinton agar plates. The test is repeated in triplicate. The number of colony forming units (CFU/mL) was determined and averaged after overnight incubation of the plates at 37 °C [27].

### 2.9. Antimicrobial Efficacy and Durability of Silver Killed Bacteria And Supernatant Against the Tested Strains

Silver-killed *E. coli* O104:H4 bacteria and supernatant prepared using AgNO_3_ at a concentration of 6 ppm were tested for their efficacy and durability against *E. coli* O104:H4, *E. coli* O157:H7, MDR *Ps. aeruginosa* and MRSA. The tested agents were prepared and stored in dark at 4 °C. The test is repeated in triplicate (zones of inhibition are averaged). First, baseline line zone was determined for each of the tested agents after their preparation. Then, the tested agents were stored in dark and tested for their antimicrobial activity after 1, 2, 6, 16, 22, 34, 40 days using agar well diffusion method. Zones of inhibition were measure and recorded as diameters in mm [28].

### 2.10. Statistical Analysis

Each experiment was done in triplicate. Data was represented as mean ± SD. One-Way ANOVA was employed to evaluate any significant difference between values obtained without the tested agents (controls) and those observed in the presence of silver-killed bacteria and supernatant. Differences were done using SPSS, 17 statistical software (SPSS Inc., Chicago, IL, USA).

## 3. Results and Discussion

Silver has long been used as antimicrobial agent in the treatment of infections, dating back to the 19th century. Specifically, silver nitrate has shown different effects against bacteria as at high concentrations, killing bacteria by different mechanisms, which are: binding to the thiol groups of protein and denaturing them, programmed cell death (apoptosis) and causing the DNA to be in the condensed form (not in the relaxed form), which inhibits cell replication. While at low concentrations, bacteria can synthesis silver nanoparticles [29]. With the nanoparticles, the size has been determined by electron microscopy; however, it is recognized that any further study will require additional assessment using a particle size analyzer.

In relation to the experimental methodology, the large inhibition zones obtained for the pellet and the supernatant against *E. coli* O104:H4 are shown in Figure 1 (inhibition zones (25 ± 0.22 mm for pellet suspension and 27 ± 0.41 for supernatant).

To demonstrate that the antibacterial activity of the pellet was due to the bacterial residue adsorbing silver and not due to toxins or enzymes produced by the bacterial cells, the effect of heat-killed bacteria (killed by autoclaving) on viable bacteria was examined. Heat-killed bacteria had no effect on the viable bacteria which suggests that silver-killed bacteria are reservoirs of silver (Figure 2). This supports a study conducted by Wakshlak, et al. [25].

The killing activity of both killed bacteria and its supernatant was shown to increase with an increase in silver concentration. We noticed that at first, supernatant showed a killing activity lower than the effect of killed bacteria at concentrations of 1.5 and 3 ppm. Then, the killing activity of both killed bacteria and its supernatant tend to be close at the concentration of 6 ppm followed by increasing in the activity of the supernatant to a greater extent than the killing activity of killed bacteria at concentrations of 12 and 18 ppm (Figure 3). These findings were close to that obtained by Wakshlak and colleagues who explained these activities at low concentrations of silver nitrate as arising from the silver metal being chelated within the bacterial cell [25]. Chelation of silver within bacterial cell has a certain limit; the excess will be available in the supernatant increasing its killing activity against the tested microorganism, as chelation becomes limited and supernatant also shows biocidal activity.

We used killed bacteria and supernatant, where their killing activities were approximately equal when tested against *E. coli* O157:H7, MDR *Ps. aeruginosa* and MRSA. It was found that a pellet suspended in deionized water and supernatant showed inhibition zones diameters that ranged from 16± 0.4 to 30 ± 0.23 mm. Also, we noticed that there are no significant differences in inhibition zone diameters obtained from neither killed bacteria nor its supernatant towards any of the tested organisms *p* > 0.05 (Appendix A & Figure 4). The activity of both killed bacteria and supernatant towards other microorganisms may be attributed to the fact that AgNps generate reactive oxygen species which have cytotoxic effect on the treated bacteria [30,31]. In addition, the presence of fresh viable bacteria acts as new, unoccupied site for silver, and so silver is shifted, according to Le-Chattelier principle, from the dead bacteria to the fresh ones [25]. In addition, when silver comes into contact with bacteria, AgNPs tends to accumulate at the bacterial membrane and form aggregates [32,33]. Other studies had supported this finding and found that the amount of adsorbed Ag+ ions and their aggregation as silver nanoparticles affects the antibacterial efficacy. As nanoparticles release ions, unless they consist of fully insoluble compounds, [34] and as a result, silver is released from nanoparticles by slow oxidation [35]. The release of silver depends on the size of the nanoparticles, their surface area, the temperature, and the composition of the surrounding medium [36,37]. In keeping with this, Xiu, et al. [38] found that in absence of dissolved molecular oxygen, there is no dissolution of silver nanoparticles.

TEM images of bacteria treated with silver nitrate clearly showed the accumulated silver as small nanoparticles having larger surface area distributed throughout the bacterium’s cross section and released outside the cell. Silver nanoparticles not only adhered at the surface of cell membrane but also penetrated inside the bacterial cell (Figure 5). Kooti, et al. [39] reported that silver had a significant effect on bacterial cell wall, changing the shape of cells from rods to cocci or irregular shapes, causing loss of cell wall integrity, releasing of cytoplasmic content, triggering swelling of bacterial cell, and finally cell lysis. Le Ouay and Stellacci [40] reported that nanoparticles that get close to a bacterium can release several tens of thousands of silver atoms in the organism’s vicinity, producing a locally high concentration of antibacterial ions. This is known as the Trojan horse effect; and here adhesion and bioactivity of positively charged Ag^+^ towards the negatively charged bacterial cell is due to electrostatic forces [41]. Silver nanoparticles, when penetrating the bacteria, interact with its DNA [2], inactivate its enzymes, generate hydrogen peroxide, causing bacterial cell death. In addition, the particles bind to functional groups of proteins, resulting in protein denaturation [10].

A time kill assay was performed to test the effect of the tested agents on bacterial viability and to assess the time necessary to reach the bactericidal activity threshold. Figure 6 showed that pellet suspension and the supernatant caused a sudden drop in the average number of viable cells showing no colonies or growth after one minute for *E. coli* O104:H4 and after 15 min for *Ps. aeruginosa* and MRSA. On the other hand, the bactericidal action of silver nitrate solution at a concentration of 6 ppm was observed after 30 min. For *E. coli* O157:H7, pellet suspension and the supernatant showed bactericidal activity after 30 min with no detectable CFUs; while silver nitrate solution achieved the bactericidal effect against *E. coli* O157:H7 after 1 h. Das, et al. [42] found that bactericidal activity of AgNO_3_ was achieved after 8 h at MBC concentration against *E. coli* and *S. aureus*. Also, the researchers reported that silver nanoparticles were effective in inhibiting bacterial growth and reproduction in a dose and time dependent manner, which agrees with our study. In another study conducted by Yamanaka, et al. [43], AgNO_3_ was shown to exhibit the bactericidal activity after 14 h. Pal, et al. [44] reported that the effect of silver nanoparticles was similar to that of the silver ions; in contrast, our results showed that pellet suspension containing the formed silver nanoparticle had higher activity than the tested silver nitrate.

Figure 7A,B showed the result of the efficacy and the durability of pellet (killed bacteria) suspension and the supernatant, against the tested bacteria. The baseline zones of inhibition diameters were determined after the preparation of the tested agents. Then, pellet suspension and supernatant were stored at 4 °C in dark. We tested the agent for its antibacterial activity at 1, 2, 6, 16, 22, 34 and 40 days. Results showed that pellet suspension and the supernatant continued to exhibit antibacterial activity until 40 days. Furthermore, the antibacterial activity of silver-killed *E.coli* O104:H4 cells and its supernatant was nearly the same against all the tested strains (no significant difference between the average zone of inhibition diameters produced by pellet suspension and the supernatant against each strain *p* > 0.05) after 24 h, 48 h, 6 days and 16 days against *E.coli* O157:H7, MRSA and MDR *P. aeruginosa* isolates, indicating the sustained release of silver ions from the silver-killed cells and its supernatant up to 40 days. While nanotoxicology research remain on-going, current data suggests there is not an on-going risk to human health once this time period has elapsed [14,34,45].

Thus, small dose of silver is enough to kill high number of bacteria and it was observed that this element is released in a gradual and controlled manner to provide an adequate amount of the antimicrobial activity for an extended period of time. Testing both as soluble silver salts release silver ions when they come in contact with water and these silver ions are the biochemically active agent [46] which suggests its use in water disinfection.

Guo, et al. [45] reported that the slower release of silver ions helps to prevent rapid depletion of the film. Such a slow and continuous release of Ag^+^ ions was due to the adhesion forces between coatings and the substrates ensuring a lasting antibacterial performance. These researchers also investigated the different bond strength between anchored Ag^+^ and electron donor (e.g., O, N and S), which resulted in different Ag^+^ release rate. Another study showed a high release of silver from Ag NPs at the first day (0.35 μg/mL), inferring that Ag NPs can quickly prevent bacterial invasion when covering a wound site. Furthermore, constant silver ion release could still be observed at the seventh day, indicating that Ag NPs had prolonged and steady antibacterial activity, which could protect cutaneous wounds from infection [47]. In addition, a further study showed that a sustained-release silver dressing had faster wound healing and lower levels of pain for the patient, facilitating earlier hospital discharge. Moreover, this helped to reduce expenses for a burns-treating medical facility [48]. Therefore, modern silver dressings are set to become very important in coming years for burn treatment; thus, understanding the clinical efficacy coupled with the cost effectiveness will become a very important subject in wound management.

## 4. Conclusions

Silver-killed bacteria can act as effective weapons against viable bacteria which are present in their environment or media. This is, as literature suggests, by acting as a reservoir for adsorbed silver nanoparticles. This phenomenon may play an important role in the management of wound infections using dressings containing silver and for the disinfection of water using silver.

## Figures and Tables

**Figure 1 antibiotics-09-00181-f001:**
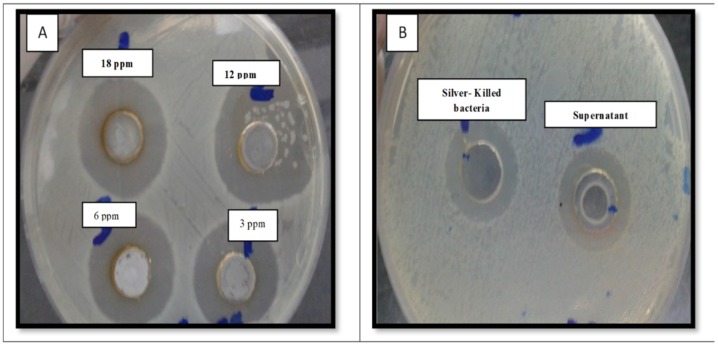
Antimicrobial activity of silver nitrate solution of different concentration (**A**), silver-killed bacteria and supernatant (**B**) against fresh viable *E. coli* O104:H4 (inhibition zones (25 ± 0.22 mm for pellet suspension and 27 ± 0.41 for supernatant).

**Figure 2 antibiotics-09-00181-f002:**
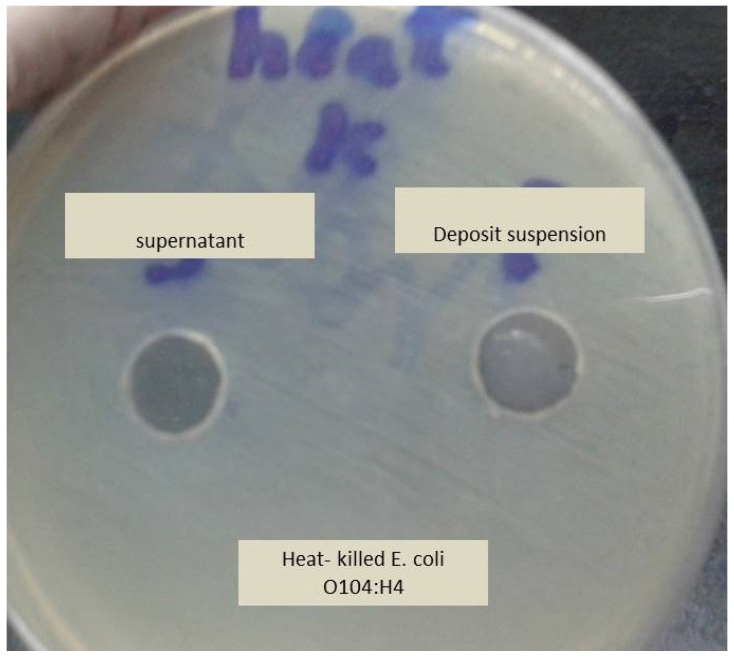
The effect of heat-killed bacteria (autoclaved) on the growth of fresh viable *E. coli* O104:H4. No antimicrobial activity was shown on the bacterial growth.

**Figure 3 antibiotics-09-00181-f003:**
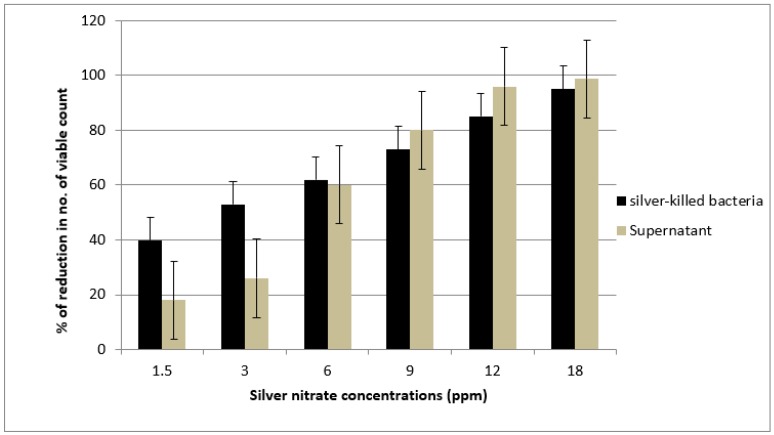
The antimicrobial activity of silver treated *E. coli* O104:H4 (killed bacteria and supernatant solution) against fresh viable bacterial culture after 6 h exposure.

**Figure 4 antibiotics-09-00181-f004:**
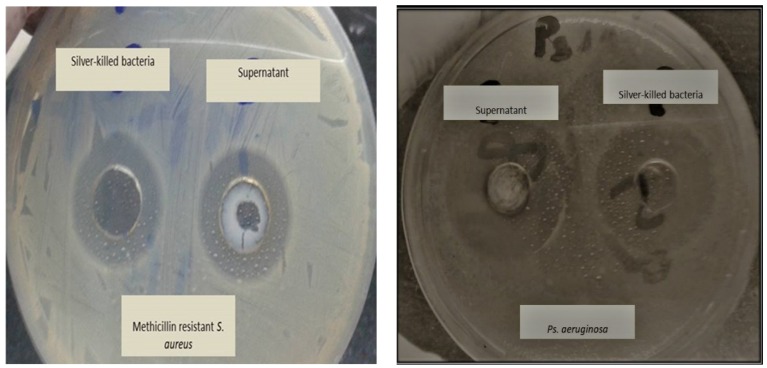
Antimicrobial activity of Silver treated bacteria (killed bacteria and the supernatant) on *Ps. aeruginosa* and Methicillin resistant *Staph. aureus* (MRSA) using well agar diffusion method. No significant difference between inhibition zone diameters obtained by killed bacteria suspension and supernatant.

**Figure 5 antibiotics-09-00181-f005:**
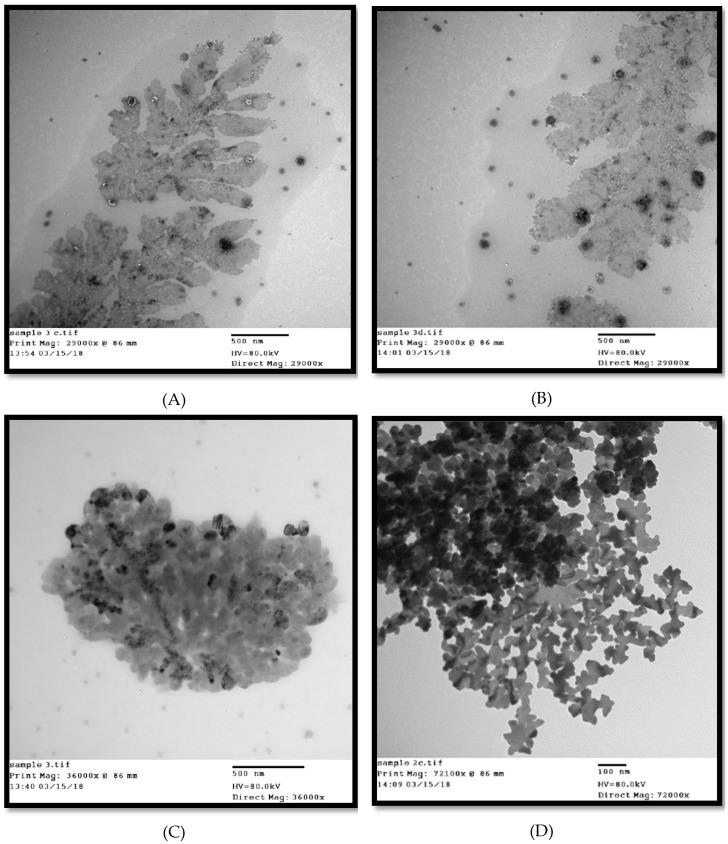
Transmission electron microscope (TEM) micrograph of *E. coli* treated with silver nitrate showing aggregation to the bacterial cells with disfigured and irregular membranes. Images showed the adsorption of AgNPs on the surface of the bacterial cells and their penetration causing cells damage (**A**,**B**,**C**, and **D**). Images A and B showed AgNPs released from the cell to the surrounding environment.

**Figure 6 antibiotics-09-00181-f006:**
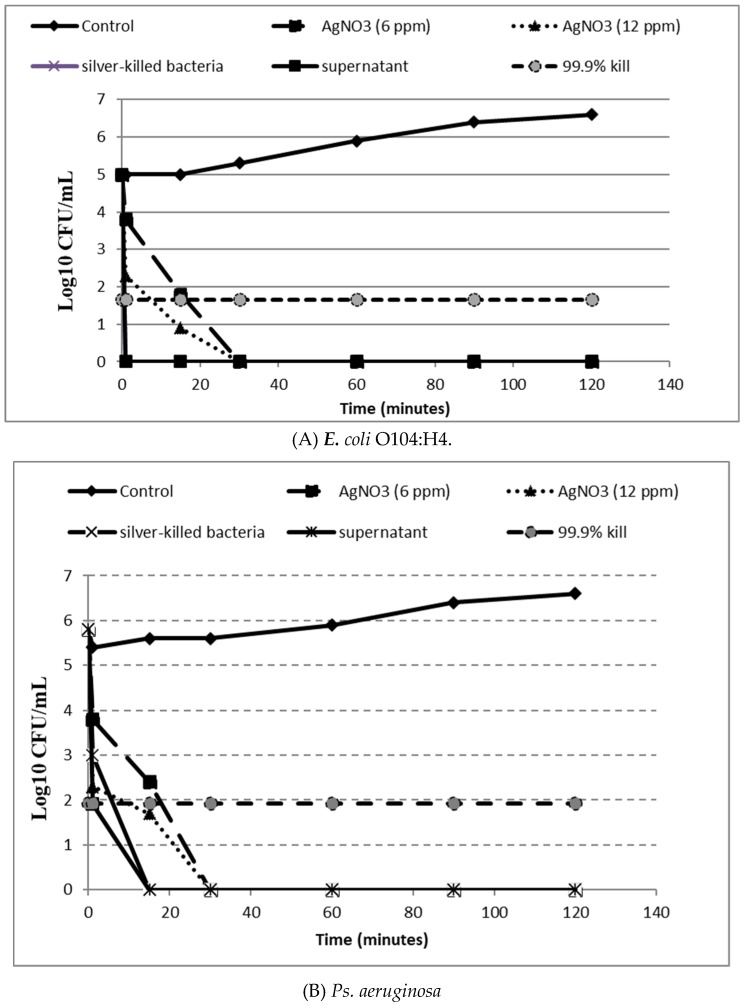
Time-kill curves of silver treated bacteria (killed bacteria suspension and supernatant), silver nitrate solutions at 6 and 12 ppm against *E. coli* O104:H4 (**A**) *Ps. aeruginosa* (**B**) MRSA (**C**) *E. coli* O157:H7 (**D**) in comparison to control (untreated).

**Figure 7 antibiotics-09-00181-f007:**
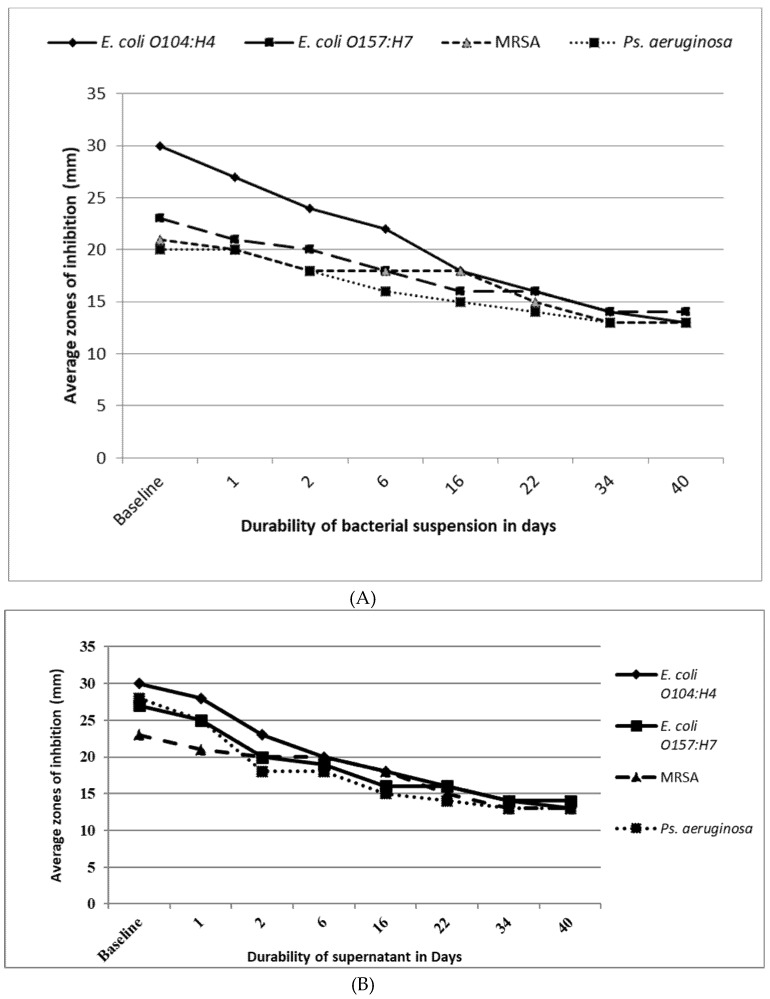
Baseline antimicrobial activity, efficacy and durability of silver treated *E. coli* O104:H4 (killed- bacteria suspension (**A**) and supernatant (**B**)).

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
