# Peer review of "Antimicrobial Activity of Silver-Treated Bacteria against other Multi-Drug Resistant Pathogens in Their Environment"

_antibiotics, 2020, doi:10.3390/antibiotics9040181_

Round 1

Reviewer 1 Report

This is an interesting study focusing on the antimicrobial activity of silver-treated bacteria against different pathogens which can be used for the management of wound infections. Although this manuscript discussed the findings very well by refereeing the existing research, it lacks proper data presentation. Some minor issues related to its presentation and clarity need to be fixed to make this work publishable.

General issues:

-In this manuscript the term “nanoparticles” was used as general without any experimental proof except the TEM micrograph. So to claim as silver NPs need the suitable experimental proof e.g. using particle size analyzer.

-Check and remove the typographical errors

Specific issues:

Introduction:

Page 1, lines 6-8, what are the other ways to prolong antimicrobial activity, please mention briefly with advantages and disadvantages. Then bring the approach you used with the benefits over other approaches.

Page 2, last line of Intro, need justification of the selection of the pathogens author used?

Materials and methods:

Page 3, lines 5-7, what was the basis of adding 1 ml bacterial suspension with 3 ml AgNO3?

Page 3 subsection 2.8, why only 6 ppm concentration was used for the in-vitro time kill assay?

Page 3, subsection 2.9, why only 6 ppm concentration was used for this test?

Page 3, subsection 2.9, line 5, how did you select these time points for this test?

Results and Discussion:

Second Paragraph is a repeated part of method. Just mention which is necessary for this section.

Figure 1 better to show as a table but still you can keep this figure as supplementary

Page 5, Lines 6-8 “These findings were close to that obtained by who explained these activities at low concentrations of silver nitrate as arising from the silver metal being chelated within the bacterial cell”, need a reference.

Figure 4 should go as supplementary and data needs to present as table

Page 11, Lines 6-9, since it showing release up to 40 days, need discussion about the safety/toxicity of silver particles if it stays 40 days in the body?

Figure 7 needs to put A and B for better identification which is mentioned in the text. Also need consistency at legend of 7A and 7B while in 7B all are as numbers (1, 2……40), 7A is (one, 2…..40).

In the last sentence of this section delete “is a very”

Conclusions:

Second sentence “This is by acting as a reservoir for adsorbed silver nanoparticles”, is it really shown in the text as a clear experimental finding or just from the literature.

Author Response

Reviewer 1

Open Review

English language and style

( ) Extensive editing of English language and style required 
( ) Moderate English changes required 
(x) English language and style are fine/minor spell check required 
( ) I don't feel qualified to judge about the English language and style 

Yes

Can be improved

Must be improved

Not applicable

Does the introduction provide sufficient background and include all relevant references?

( )

(x)

( )

( )

Is the research design appropriate?

(x)

( )

( )

( )

Are the methods adequately described?

( )

(x)

( )

( )

Are the results clearly presented?

( )

( )

(x)

( )

Are the conclusions supported by the results?

( )

(x)

( )

( )

Comments and Suggestions for Authors

This is an interesting study focusing on the antimicrobial activity of silver-treated bacteria against different pathogens which can be used for the management of wound infections. Although this manuscript discussed the findings very well by refereeing the existing research, it lacks proper data presentation. Some minor issues related to its presentation and clarity need to be fixed to make this work publishable.

English language was revised and edited.

General issues:

-In this manuscript the term “nanoparticles” was used as general without any experimental proof except the TEM micrograph. So to claim as silver NPs need the suitable experimental proof e.g. using particle size analyzer.

With the nanoparticles, the size has been determined by electron microscopy; however, it is recognized that any further study will require additional assessment using a particle size analyzer.

-Check and remove the typographical errors

 It was corrected

Specific issues:

Introduction:

Page 1, lines 6-8, what are the other ways to prolong antimicrobial activity, please mention briefly with advantages and disadvantages. Then bring the approach you used with the benefits over other approaches.

 A brief was added on the other methods prolonging antimicrobial activity in the introduction section in page 1 & 2.

Page 2, last line of Intro, need justification of the selection of the pathogens author used?

Justification was added in the last paragraph of introduction section.

Materials and methods:

Page 3, lines 5-7, what was the basis of adding 1 ml bacterial suspension with 3 ml AgNO3?

We mean that we add one ml of bacteria to AgNO3 with a total volume of 3ml to obtain the tested concentrations as we prepare concentrations of 2.25, 4.5, 9, 18 and 27 ppm. So, we need to add 2 ml of silver nitrate to on ml of bacteria to give the final concentrations of 1.5, 3, 6, 12 and 18 ppm. We did modification in the experiment to clear our point. We wrote this part as follows: The bacterial suspension (1.0 ml) was added to sterile centrifuge tubes containing 2 ml solution ofAgNO3 3.0 ml total volumeat increasing final concentrations of AgNO3 (1.5, 3, 6, 12, 18 ppm) and the mixture was incubated overnight at 37oC under dark conditions for 6 hrs

Page 3 subsection 2.8, why only 6 ppm concentration was used for the in-vitro time kill assay? And Page 3, subsection 2.9, why only 6 ppm concentration was used for this test?

In page 5, in the section of results and discussion we addressed that the killing activity of killed bacteria at concentration of 6 ppm were nearly similar. So, we choose this concentration to compare between the effect killed bacterial and supernatant. As, We noticed that at first, supernatant showed a killing activity lower than the effect of killed bacteria at concentrations of 1.5 and 3 ppm. Then, the killing activity of both killed bacteria and its supernatant tend to be close at the concentration of 6 ppm followed by increasing in the activity of the supernatant to a greater extent than the killing activity of killed bacteria at concentrations of 12 and 18 ppm which may be explained as follow: at low concentrations, silver nitrate activities arises from the silver metal being chelated within the bacterial cell. Chelation of silver within bacterial cell has certain limit; the excess will be available in the supernatant increasing its killing activity against the tested microorganism. As chelation becomes limited and supernatant also shows biocidal activity.

Page 3, subsection 2.9, line 5, how did you select these time points for this test?

Previously upon performing our experiment at different time interval, we noticed significant observable changes and effects at these time intervals. So, we decided to write this time interval.

Results and Discussion:

Second Paragraph is a repeated part of method. Just mention which is necessary for this section.

It was modified.

Figure 1 better to show as a table but still you can keep this figure as supplementary

We see that presentation of the results as figures in this section ensures the activity of the tested agents but if you see that it necessary to be added as supplementary, we will do that.

Page 5, Lines 6-8 “These findings were close to that obtained by who explained these activities at low concentrations of silver nitrate as arising from the silver metal being chelated within the bacterial cell”, need a reference.

Wakshlak, R. B., R. Pedahzur, and D. Avnir. "Antibacterial Activity of Silver-Killed Bacteria: The "Zombies" Effect." Sci Rep 5 (2015): 9555.

Figure 4 should go as supplementary and data needs to present as table

We represented results as a table as a supplementary data

Page 11, Lines 6-9, since it showing release up to 40 days, need discussion about the safety/toxicity of silver particles if it stays 40 days in the body?

While nanotoxicology research remain on-going, current data suggests there is not an on-going risk to human health once this time period has elapsed.  Ag+ concentration in mammalian cells shouldn’t exceed toxicity potential dose range which is between 1 and 25 ppm and the amount of orally taken Ag NPs is from 0.4 to 27 μg per day. In addition, we recommend it as a good water disinfectant.

Figure 7 needs to put A and B for better identification which is mentioned in the text. Also need consistency at legend of 7A and 7B while in 7B all are as numbers (1, 2……40), 7A is (one, 2…..40).

It was performed

In the last sentence of this section delete “is a very”

It was deleted

Conclusions:

Second sentence “This is by acting as a reservoir for adsorbed silver nanoparticles”, is it really shown in the text as a clear experimental finding or just from the literature.

From previous literature and the overall results of our work (see conclusion).

Reviewer 2 Report

  1. Introduction – 3rd paragraph
  • “Because of these properties, silver is a practical antimicrobial agent which has been can be utilized in years to come. “
  1. Why use coli O104:H4 - Shiga toxin–producing Escherichia coli?

  1. "All cultures were maintained in their appropriate agar slants at 4°C and used as stock cultures."
  • Is it possible that the maintenance temp. could have affected the viability of the bacterial isolates especially Escherichia coli O104:H4?
  • Figures 1 - 5: are not noticeably clear; needs formatting … too stretched out
  • Titles of the tables need modifications and rearrangement
  1. Consider separating Results from Discussion for clarity

Author Response

Reviewer 2

Open Review

English language and style

( ) Extensive editing of English language and style required 
( ) Moderate English changes required 
(x) English language and style are fine/minor spell check required 
( ) I don't feel qualified to judge about the English language and style 

Yes

Can be improved

Must be improved

Not applicable

Does the introduction provide sufficient background and include all relevant references?

(x)

( )

( )

( )

Is the research design appropriate?

(x)

( )

( )

( )

Are the methods adequately described?

(x)

( )

( )

( )

Are the results clearly presented?

( )

(x)

( )

( )

Are the conclusions supported by the results?

( )

(x)

( )

( )

Comments and Suggestions for Authors

Introduction – 3rd paragraph     

- Because of these properties, silver is a practical antimicrobial agent which has beencan be utilized in years to come.

It was modified in the introduction section.

- Why use coli O104:H4 - Shiga toxin–producing Escherichia coli?

coli O104:H4 is an important water and food-borne pathogen that combines the virulence characteristics of enterohemorrhagic E. coli(EHEC) and enteroaggregative E. coli (EAEC) and can cause severe diseases and outbreaks.Ps. aeruginosa and S. aureus were selected because these pathogens represent common causes of wound, burn and nosocomial infectious agents; E. coli 0157: H7 was selected as a representative strain of food-borne pathogen (and it is of clinical significance).

- "All cultures were maintained in their appropriate agar slants at 4°C and used as stock cultures."

It was modified

- Is it possible that the maintenance temp. could have affected the viability of the bacterial isolates especially Escherichia coli O104:H4?

- This temperature regime ensures that the bacteria used did not grow in number while maintaining cell viability (viability was demonstrated using controls). It is added to 2.2 section.

- Figures 1 - 5: are not noticeably clear; needs formatting … too stretched out

It was modified

- Titles of the tables need modifications and rearrangement

It was revised

-Consider separating Results from Discussion for clarity.

We wrote these combined to avoid the repetition of results and to be able to explain all results in detail.

English language was revised and edited.

Reviewer 3 Report

This paper provides important data on the antimicrobial activity of silver-treated bacteria against notable multi-drug resistant pathogens. I personally hope this paper will be accepted for publication after considering suggestions described below.

1) Many simple typographical errors were found in this manuscript. Please check carefully again before next submission. ex) AgNO“3”; pages 6 & 11, P “>” 0.05 is correct?; especially page 3, writing style is not uniformed. “E. coli” and “E. coli” / 24 “hr”, 6 “hrs”, 24 “h”, etc.

2) Figure 2 seems to be enlarged vertically. Please try to consider modifying it. In addition, I think it is better to add scale (mm) to the pictures.

3) I think Figure 4 on the right side is dark and hard to see. Please try to make it clear.

4) In Figure 5, what is the exact difference among three pictures (C, D and E)? I think two pictures (C & D or D & E, etc.) are enough.

5) Did the authors also perform in-vitro time kill assay (2.8.) & antimicrobial efficacy and durability test (2.9) in triplicate (n ≥ 3)? If so, figures 6 & 7 show each average?

6) I think Acknowledgments part is the same as that of Funding part. I suggest the authors should remove or rewrite again.

I hope my comments are useful for the improvement of this paper.

Author Response

Reviewer 3

Open Review

English language and style

( ) Extensive editing of English language and style required 
( ) Moderate English changes required 
(x) English language and style are fine/minor spell check required 
( ) I don't feel qualified to judge about the English language and style 

Yes

Can be improved

Must be improved

Not applicable

Does the introduction provide sufficient background and include all relevant references?

(x)

( )

( )

( )

Is the research design appropriate?

(x)

( )

( )

( )

Are the methods adequately described?

(x)

( )

( )

( )

Are the results clearly presented?

(x)

( )

( )

( )

Are the conclusions supported by the results?

(x)

( )

( )

( )

Comments and Suggestions for Authors

This paper provides important data on the antimicrobial activity of silver-treated bacteria against notable multi-drug resistant pathogens. I personally hope this paper will be accepted for publication after considering suggestions described below.

1. Many simple typographical errors were found in this manuscript. Please check carefully again before next submission. ex) AgNO“3”; pages 6 & 11, P “>” 0.05 is correct?; especially page 3, writing style is not uniformed. “ coli” and “E. coli” / 24 “hr”, 6 “hrs”, 24 “h”, etc.

All typographical errors were corrected

2) Figure 2 seems to be enlarged vertically. Please try to consider modifying it. In addition, I think it is better to add scale (mm) to the pictures.

It was modified

3) I think Figure 4 on the right side is dark and hard to see. Please try to make it clear.

We tried to make it clear

4) In Figure 5, what is the exact difference among three pictures (C, D and E)? I think two pictures (C & D or D & E, etc.) are enough.

Image E was deleted

5) Did the authors also perform in-vitro time kill assay (2.8.) & antimicrobial efficacy and durability test (2.9) in triplicate (n ≥ 3)? If so, figures 6 & 7 show each average?

Section 2.8 and 2.9 were modified.

6) I think Acknowledgments part is the same as that of Funding part. I suggest the authors should remove or rewrite again.

It was revised

I hope my comments are useful for the improvement of this paper.